# Nanotechnology-Enhanced Cosmetic Application of Kojic Acid Dipalmitate, a Kojic Acid Derivate with Improved Properties

Angreni Ayuhastuti [1], Insan Sunan Kurniawan Syah [2], Sandra Megantara [3] and Anis Yohana Chaerunisaa [2,*]

1   Doctoral Study Program, Faculty of Pharmacy, Universitas Padjadjaran, Jl. Raya Jatinangor Km 21.5, Sumedang 45363, West Java, Indonesia; angreni22001@mail.unpad.ac.id
2   Department of Pharmaceutics and Pharmaceutical Technology, Faculty of Pharmacy, Universitas Padjadjaran, Sumedang 45363, West Java, Indonesia; insan.sunan@unpad.ac.id
3   Department of Pharmaceutical Analysis and Medicinal Chemistry, Universitas Padjadjaran, Sumedang 45363, West Java, Indonesia; s.megantara@unpad.ac.id
*   Correspondence: anis.yohana.chaerunisaa@unpad.ac.id

**Abstract:** Kojic acid (KA) has emerged as a prominent tyrosinase inhibitor with considerable potential in cosmetic applications; however, its susceptibility to instability during storage poses a challenge to its widespread use. This review explores the advancements in addressing this limitation through the development of various KA derivatives, focusing on the modification of the C-7 hydroxyl group. Strategies such as esterification, hydroxy-phenyl ether formation, glycosylation, and incorporation into amino acid or tripeptide derivatives have been employed to enhance stability and efficacy. Among these derivatives, Kojic Acid Dipalmitate (KDP), a palmitic ester derivative of KA, stands out for its notable improvements in stability, permeability, and low toxicity. Recent developments indicate a growing utilization of KDP in cosmetic formulations, with over 132 available products on the market, encompassing various formulations. Formulations based on nanotechnology, which incorporate KDP, have been provided, including nanosomes, nanocreams, multiple emulsions, liposomes, solid lipid nanoparticles (SLNs), ethosomes, and nanoemulsions. Additionally, three patents and seven advanced system deliveries of KDP further underscore its significance. Despite its increasing prevalence, the literature on KDP remains limited. This review aims to bridge this gap by providing insights into the synthesis process, physicochemical properties, pharmaceutical preparation, diverse applications of KDP in cosmetic products, and recent nanotechnology formulations of KDP. This review paper seeks to explore the recent developments in the use of KDP in cosmetics. The goal is to enhance stability, permeability, and reduce the toxicity of KA, with the intention of promoting future research in this promising sector.

**Keywords:** kojic acid dipalmitate; cosmeceuticals

## 1. Introduction

Tyrosinase (monophenol, L-dopa: oxygen oxidoreductase, EC 1.14.18.1) is known as the starting point for the formation of mammalian skin color [1]. It is an enzyme that catalyzes several steps in the production of the pigment melanin in living cells, including bacteria, fungi, plants, animals, and humans [1,2]. It is located in melanocytes in the epidermis, especially in the viable epidermis [3,4]. Tyrosinase is the enzyme that controls the pace of melanin synthesis [5,6] which is the process responsible for producing the pigment that determines skin color [7].

Melanin synthesis, or melanogenesis, is a complex process that involves various protein groups, including tyrosinase, tyrosinase-related protein 1 (Tyrp1 or TRP1), and tyrosinase-related protein 2 (Tyrp2, DCT, or TRP2) [6–8]. Melanogenesis occurs in an auto-regulated manner. The activity of tyrosinase begins with the presence of the substrate tyrosine and the enzyme co-factor, dihydroxyphenylalanine (DOPA). Tyrosinase uses its binuclear copper center to hydroxylate tyrosine into 3,4-dihydroxyphenylalanine

(DOPA) [9,10]. Then, tyrosinase catalyzes the oxidation of DOPA to DOPAquinone [11]. This reaction proceeds with multi-polymerization to form pigments that are blackish-gray in color, namely eumelanin, and red-yellowish in color, namely pheomelanin, with the influence of the conjugation reaction [12].

This melanogenesis process occurs in the melanosome [13], where the size, density, and shape of the melanosome among populations have the same characteristics [14,15]. The determinant of skin color for a population is the total amount, ratio, and distribution of eumelanin and pheomelanin, which differ among populations around the world, such as Europe, Africa, and Asia [16]. In some countries, particularly in Southeast Asia, a high amount of eumelanin is undesirable because fair and clean skin has become the standard of beauty for women in these countries [14,17]. This is evident from a study by Peltzer (2016) of 19,624 students from 26 low-, middle-, and developing-income countries, showing that Southeast Asia has a higher percentage of skin-lightening product users than Africa, at 36.0% [18]. Indonesia, with a majority of Fitzpatrick skin phototypes IV and V, which tend to be dark or brown [18], has a skin-lightening product usage rate of up to 36% [19]. Abnormal skin pigmentation in the form of hypo- or hyperpigmentation can cause significant anxiety and decrease self-esteem in affected individuals [18,19]. Various methods are employed to regulate pigmentation in the fields of dermatology and cosmetics. One of these methods involves the utilization of synthetic compounds, such as hydroquinone and kojic acid (KA) [20–22].

Hydroquinone is a gold standard compound for treating hyperpigmentation [20,23,24]. However, its use in cosmetic formulations is prohibited due to the side effects such as irritation, allergic reactions, post-inflammatory hyperpigmentation, and temporary hypopigmentation that it can cause [12,13,15–18,25,26]. Hydroquinone ($C_6H_6O_2$) reduces the level of pigmentation by non-selectively degrading epidermal melanocytes and keratinocytes, making it cytotoxic to cells [27,28]. Therefore, the use of skin-lightening agents in cosmetic formulations has shifted towards more effective alternative compounds with low toxic and irritation effects, such as kojic acid [23,29].

Kojic acid is one of several tyrosinase inhibitors that have been extensively studied for this purpose [22,30–33]. It is a natural compound with both skin-lightening and antibacterial properties and is widely used for cosmetic purposes and as a food additive to prevent browning caused by enzymes [20,23,24]. While KA has a competitive inhibitory effect on the monophenolase activity and a mixed inhibitory effect on the diphenolase activity of mushroom tyrosinase, its use in cosmetics is limited by its instability during storage due to its labile oxidative properties, which can be accelerated by light and heat [34,35]. To address these limitations, various KA derivatives have been developed by modifying the C-7 hydroxyl group, such as through esterification [36], hydroxyphenyl ether formation [37], glycosylation [38], or incorporation into amino acid or tripeptide derivatives [39], with the aim of improving their stability and efficacy in cosmetic and cosmeceutical applications.

According to reports, kojic acid–tripeptide amide derivatives have shown superior storage stability in comparison with kojic acid [40]. Additionally, as stated in Rho et al. (2010) [41], kojyl thioether derivatives strongly inhibit tyrosinase activity. Moreover, Lee et al. (2006) [38] report that kojic acid derivatives with two pyrone rings possess eight times higher tyrosinase inhibitory potency than kojic acid itself. Maltol (3-hydroxy-4H-pyran-4-one) and its derivatives share a similar scaffold with kojic acid and have similar biological effects. Ester derivatives of allomaltol (5-hydroxy-2-methyl-4H-pyran-4-one) have been described to have inhibitory to tyrosinase and antioxidant effects by Wempe and Michael (2012) [42]. Kojic acid has also been reported to exhibit antioxidant activity [43]. According to Ahn el al. (2011), a kojic acid derivative containing a trolox moiety exhibits potent tyrosinase inhibitory and radical scavenging activity [44]. Lajis et al. (2012) suggest that KA esters derived from the esterification of kojic acid and palm oil-based fatty acids, namely, kojic acid monooleate, kojic acid monolaurate, and kojic acid monopalmitate, exhibit similar inhibitory effects to kojic acid; however, kojic acid monopalmitate gave slightly stronger inhibition to melanin formation compared with other inhibitors [45]. Moreover, Balaguer

et al. (2008) reported that kojic acid dipalmitate (KDP) poses superior stability, oil solubility, and skin absorption compared with kojic acid, attributed to its resistance to changes in pH, heat, and light compared with kojic acid [34].

Kojic Acid Dipalmitate (KDP), a palmitic ester derivative of KA, has gained widespread usage in cosmetic formulations in recent times due to its improvement in stability and permeability, as well as its low toxicity [21,26,46]. It is synthesized in skin cells through an in situ esterification process, which results in the release of kojic acid [47]. This unique characteristic sets it apart from other derivatives of kojic acid [48]. The aforementioned condition has resulted in the widespread usage of KDP, which is a commonly utilized component in numerous skincare items, including creams and serums, by well-known cosmetic brands. These products usually contain concentrations of up to 3% KDP and are marketed as skin-whitening and lightening agents [49]. Nanotechnology-based formulations containing Kojic Acid Dipalmitate (KDP) have been developed, including nanosomes, nanocreams, multiple emulsions, liposomes, solid lipid nanoparticles (SLNs), ethosomes, and nanoemulsions. The primary objective of these formulations is to enhance the penetration of KDP into melanosomes.

Despite its widespread use, the available literature on KDP remains limited. This review aims to discuss various aspects of KDP, including its synthesis process, physicochemical properties, pharmaceutical preparation, and application in cosmetic products. Although KDP is becoming more common, there is still a scarcity of literature on the subject. This thorough review seeks to fill this need by offering insights into the synthesis process, physicochemical properties, pharmaceutical preparation, diverse applications of KDP in cosmetic products, and recent nanotechnology formulations of KDP. This review study aims to examine the latest advancements in the use of KDP in the field of cosmetics. The objective is to improve the stability, permeability, and toxicity of KA in order to facilitate further investigation in this promising field.

## 2. Kojic Acid Derivatives

Kojic acid, scientifically 5-hydroxy-2-hydroxymethyl-4H-pyran-4one (Figure 1a), is a well-researched substance that effectively inhibits the enzyme tyrosinase [50]. Kojic acid, which was first found by Saito in 1907 in Japan, was extracted from the culture of *Aspergillus oryzae* that was growing on steam rice [51–53]. The chemical structure was determined by Yabuta in 1924 [54]. Kojic acid is an organic acid substance derived from the fermentation process of fungus, specifically from over 58 different fungal strains belonging to genera such as *Aspergilus*, *Acetobacter*, and *Penicillium* [49,55]. Currently, kojic acid is used as a food ingredient to prevent enzymatic browning [32] and in the cosmetic business as a skin-brightening agent [50,51].

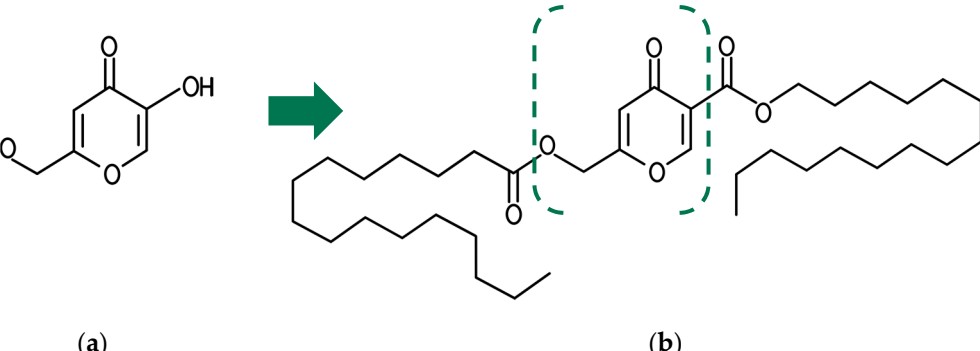

(**a**)                                                    (**b**)

**Figure 1.** The basic molecular structure of (**a**) kojic acid (KA) and (**b**) kojic acid dipalmitate (AKD). The intermittent lines indicates the location of the esterification of KA with two palmitic acids, resulting in the formation of AKD.

In the cosmetic field, the use of Kojic Acid (KA) is limited due to the potential for skin irritation, such as contact dermatitis, sensitization, redness, and erythema [9]. Additionally, KA exhibits limited inhibitory activity and instability during storage [40]. Consequently, numerous derivatives of KA have been synthesized to enhance its properties. These modifications often involve converting the C-7 hydroxyl group into esters, hydroxyphenyl ethers, glycosides, amino acid derivatives [36], or tripeptide derivatives in order to address these limitations.

Modification of the Kojic Acid (KA) structure at the C-7 hydroxyl group has been undertaken by Chen et al. [56,57]. Through this modification, compounds were synthesized that exhibited potent inhibition of mushroom tyrosinase. Among the synthesized compounds, 2-(((4-amino-5-(2-((E)-3-(2-methoxyphenyl)allylidene)hydrazinyl)-4H-1,2,4-triazol-3-yl)thio)methyl)-5-hydroxy-4H-pyran-4-one (KAD2) demonstrated the most effective inhibitory effects on diphenolase activity and monophenolase activity, with IC50 values of 7.50 µM and 20.51 µM, respectively. In another study, Zhao et al. [58] altered the 'O' at position-1 in the pyranone ring to 'NH', leading to the synthesis of a range of hydroxypyridinone–amino acid and hydroxypyridinone–dipeptide conjugates. Compound 6e exhibited superior copper reduction ability and a stronger copper chelating capacity compared to kojic acid. Asadzadeh et al. [59,60] produced 12 sets of kojic acid derivatives by altering the C-7 hydroxyl group and the aromatic substituent at the C-2 position. Furthermore, some substances demonstrated commendable anti-tyrosinase efficacy. Compound IIId had the most potent tyrosinase inhibitory action, with an IC50 value of $0.216 \pm 0.009$ mM. This finding is consistent with the in silico $\Delta G$ bind data, which showed a binding energy of $-13.24$ Kcal/mol. Shao et al. [61] synthesized new hydroxypyridinone derivatives that have an oxime ether by changing the C-7 hydroxyl group. These compounds have strong inhibitory effects on diphenolase activity and monophenolase activity, which successfully extend the shelf life of fresh-cut apple slices. Furthermore, hydroxyl group modifications were conducted by Ashooriha et al. [62] by substituting alcoholic hydroxyl groups with sixteen suitable substituents. All compounds exhibited very effective anti-tyrosinase activity, with $IC_{50}$ values ranging from 0.06 to 6.80 µM. Among them, compound 6o showed the most promising results and indicated an acceptable safety profile based on cytotoxicity studies conducted on B16 melanoma cell lines and Human Foreskin Fibroblast (HFF) cells. Moreover, Xie et al. [63] found that the compound with the structure of 5-phenyl-3-[5-hydroxy-4-pyrone-2-yl-methylmercap-to]-4-(2,4-dihydroxyl-benzylamino)-1,2,4-triazole exhibited the most potent tyrosinase inhibitory activity, with an IC50 value of $1.35 \pm 2.15$ µM.

Derivatization of kojic acid has also been accomplished through the synthesis of the dimeric structure of kojic acid, as conducted by Rho et al. The synthesis involved the use of ester, amide, and thioether linkages [41]. This research demonstrated that thioether linkage resulted in the most effective inhibition of tyrosinase compared with other linkages, as well as nitric oxide (NO) production. Lee et al. [38] synthesized kojic acid with two pyron ring linkages by ethylene, exhibiting tyrosinase inhibition activity eight times more potent ($IC_{50} = 3.63$ µM) than KA ($IC_{50} = 30.61$ µM). Furthermore, modification of KA at the 6th position of the pyranone ring for antityrosinase activity was conducted by Rho et al. [64] and the impact of this modification was investigated in this study.

Karakaya et al. [64] also contributed to the synthesis of kojic acid derivatives, producing a total of thirty Mannich bases, including seventeen novel compounds with a structure of 2-substituted-3-hydroxy-6-hydroxymethyl/chloromethyl/methyl/morpholinylmethyl/ piperidinylmethyl/pyrrolidinylmethyl-4H-pyran-4-one. Among these compounds, ten derivatives exhibited higher activity than KA, and compound 3, bearing a 3,4-dichlorobenzyl piperazine moiety, demonstrated the highest inhibitory activity. The hydroxymethyl group at the 6th position of the pyranone ring was identified as plausibly binding to copper ions on the active site of the enzyme, acting as a mushroom tyrosinase inhibitor. Additionally, Cardoso et al. [65] synthesized 14 KA derivatives from malononitrile and aromatic aldehyde using β-cyclodextrin (β-CD) as a catalyst. All derivatives exhibited conformational

affinity to the enzyme's active site, with D5 (a derivative of KA containing phenolic compounds in the benzene ring) identified as the most stable KA derivative, with a binding free energy of $-18.13$ (D5) kcal mol$^{-1}$. This suggests that these derivatives could serve as potent competitive inhibitors of the natural substrates of L-DOPA and L-tyrosine in melanogenesis.

Subsequently, the hydroxyl group of kojic acid at position C-7 was modified by including amino acids or peptides. The synthesis of amino acid derivatives of kojic acid was initially conducted by Kayahara et al. in 1990 with the aim of obtaining compounds with enhanced antibacterial potential [66]. Meanwhile, the synthesis of amino acid derivatives from kojic acid to elevate tyrosinase inhibition activity was carried out by Kobayashi et al. in 1995. It was reported that six amino acid derivatives of kojic acid had been synthesized, including N-carbobenzoxy (abbr., Z)-Ala-Kojic acid derivative, Z-Thr(OH)-Kojic acid derivative, Z-Val-Kojic acid derivative, Z-Leu-Kojic acid derivative, Z-Ile-Kojic acid derivative, and Z-Phe-Kojic acid derivative. The synthesis involved the introduction of Z-protected amino acids into the 7th position of kojic acid using 1-ethyl-3,3-dimethylaminopropyl hydrochloride (EDC). Based on the evaluation of the IC$_{50}$ values for tyrosinase inhibition [EC 1.14..18.1] obtained from mushrooms, it was revealed that all amino acid derivatives of kojic acid exhibited a higher tyrosinase inhibition potential than kojic acid, with the L-phenylalanine derivative being the strongest inhibitor, displaying an IC$_{50}$ value approximately 1/80th of that for kojic acid [36]. Unnatural amino acids, which may be found naturally or created via chemical synthesis, are extensively used in ligand design. When included in therapeutic peptidomimetics and peptide analogs, they serve as a potent tool in drug development.

In alignment with this research, Kwak et al. (2007–2010) conducted the derivatization of amino acids from KA, namely KA-AA3-AA2-AA1-NH2 with varying tripeptide arrangements. As a result, kojic acid-FWY-NH2 (FWY: Phe-Trp-Tyr) was demonstrated to be the most effective compound, displaying the highest inhibitory activity, which remained consistent over different storage times under various temperatures and pH conditions [62]. Although kojic acid–tripeptide amides (KA–FWY–NH2) exhibited a 100-fold increase in tyrosinase inhibitory activity compared with kojic acid itself, KA–FWY–NH2 showed limited inhibitory activity due to its large molecular weight, hindering its ability to penetrate the cell membrane. Consequently, Kwak (2009) synthesized a kojic acid–amino acid amide (KA–AA–NH2) library to reduce molecular weight. Kojic acid–phenylalanine amide (KA–F–NH2), despite displaying the highest tyrosinase inhibitory activity equivalent to KA–FWY–NH2 in mushroom tyrosinase inhibitory tests, did not inhibit the melanin synthetic pathway in cell systems, likely due to its poor cell-penetrating ability. Researchers explained this phenomenon in relation to the hydrophobic nature of L-phenylalanine [40,67]. Introducing synthetic amino acids into peptides has the capacity to bolster their resistance against enzyme breakdown, therefore amplifying the range of structures and biological functions shown by peptides. Kojic acid-containing amino acid derivatives provide several sites for oxidation, reduction, alkylation, acylation, and peptide-coupling reactions. These chemicals show great potential for use as tyrosinase inhibitors [54]. Nevertheless, it is important to take into account the physical characteristics, such as the molecular mass, of the derivative compounds. The compound's efficiency may be diminished due to the hindered penetration of cell membranes caused by an increase in molecular weight [67].

## 3. Sythesis of Kojic Acid Dipalmitate

The hydrophilicity of KA has limited its use in cosmetics, oily foods, and pharmaceutical products. Furthermore, concerns exist regarding its potential toxicity [49] and irritancy [44,47,68–70]. To enhance the chemical and biological attributes of KA, researchers have developed derivatives with improved properties. Several efforts, including the enzymatic esterification of KA and fatty acids to form KA esters, have been undertaken [71] to increase their hydrophobicity and expand their potential uses, such as in the cosmetic

industry. Some KA esters, like KA dipalmitate, have been brought to market for cosmetic and skin health applications [72].

Kojic acid dipalmitate can be synthesized by esterifying kojic acid with palmitic acid. Figure 1 depicts the basic molecular structure of KA and KDP. Kojic acid possesses two functional groups: a hydroxyl group (OH) at C-5 and a carboxylic group (COOH) at C-7 [73]. The esterification process involves removing the OH group and attaching the fatty acid (R) to create KA monoesters like 5-O-KA monoesters and 7-O-KA monoesters. Fatty acids have been synthesized chemically and enzymatically to link with KA at positions C-5 and C-7 [74]. Chemical esterification of KA oleate was facilitated by N,N′-dicyclohexylcarbodiimide (DCC)/4-dimethylaminopyridine (DMAP) in dichloromethane, resulting in yields of up to 80% within 24 to 48 h. However, this method necessitates the use of environmentally harmful and dangerous chemicals, requiring additional safety precautions. Other chemical esterification procedures involve numerous steps and chemicals, leading to a higher production cost for KDP [45].

Enzymatic processes for esterification of KA involve the preparation and catalysis of lipases and proteases in organic or solvent-free systems, resulting in the utilization of fewer chemicals and being more cost-effective and environmentally friendly. When used in their immobilized form, most of the enzymes can be repeatedly reused, resulting in consistent specific enzyme activity and yield during the synthesis of KA esters. The yield of enzymatically synthesized KA esters is influenced by several factors, including the type of catalytic enzyme, reaction temperature, organic solvents, KA to fatty acid ratio, metal ions, water content, and pH [45,75,76].

Various enzymes have been screened for enzymatic synthesis of KA esters, and most of them were derived from fungi and bacteria. However, the highest yields were obtained when lipase enzymes from *Candida antarctica, Pseudomonas cepacia,* and *Rhizomucor miehei* were utilized. Based on research conducted by Liu and Shaw (1998) [75], Kobayashi et al. (2001) [77], Khamaruddin et al. (2008) [78], and Ashari et al. (2009) [71], the synthesis of C-5-KA monoester using these enzymes resulted in a yield of 40–60%. The optimal temperature for KA ester synthesis is closely linked to the optimum temperature of the immobilized enzyme employed in the esterification process. For instance, lipase from Pseudomonas cepacia exhibits its optimal activity at a temperature of 50 °C [75,76]. In this context, immobilized enzymes are utilized due to their thermostability and higher catalytic activity when compared to free enzymes [79]. Furthermore, the choice of an organic solvent played a significant role in influencing the esterification process. A high ratio of KA to fatty acid esterification resulting in KA esters was attained by using specific solvents, namely acetonitrile, acetone, and chloroform, which possessed logP values of −0.33, −0.21, and 2.00, respectively [80]. To enhance the hydrophobic nature of the reaction mixture and thereby improve the efficiency of the esterification process, a co-solvent mixture was also employed [80]. Moreover, KA-to-fatty acid ratio, metal ions, water content, and pH also influence the esterification process. Lajis et al. (2013) [76] have extensively discussed the influence of these parameters on esterification. Readers are encouraged to refer directly to the literature for more details.

## 4. Physical and Chemical Propreties of Kojic Acid Dipalmitate

The molecular formula of kojic acid dipalmitate (2-Palmitoyloxymethyl-5-palmitoyloxy-pyrone) is $C_{38}H_{66}O_6$, with a molecular weight of 618.9 g/mol [81]. Kojic dipalmitic acid exhibits characteristics of a white powder [47], a melting point of 94 °C, and solubility in oil, alcohol, mineral oil, and esters. Unlike kojic acid, kojic acid dipalmitate is more stable to light, heat [82], and oxidation and does not chelate metal ions [83]. This makes it more color-stable, with a reduced likelihood of turning yellow or brown, which makes it a more popular choice for manufacturers of skin-lightening whitening creams [83]. Kojic acid dipalmitate is also considered stable over a wide range of pHs [84].

The chemical structure of kojic acid dipalmitate consists of two molecules of palmitic acid, which are saturated fatty acid, attached to the two hydroxyl groups of kojic acid. This

structure gives kojic acid dipalmitate its lipophilic (fat-loving) properties, making it more soluble in oils and fats than kojic acid itself [47,85]. These derivatives have been found to improve both the stability and solubility of kojic acid in oily cosmetic products [86]. The comparison of the physical and chemical properties of kojic acid dipalmitate and kojic acid is presented in Table 1.

**Table 1.** Physicochemical properties of kojic acid dipalmitate and kojic acid.

| Properties | Kojic Acid Dipalmitate | Kojic Acid |
| --- | --- | --- |
| Molecular weight | 618.9 g/mol | 142.11 g/mol |
| Solubility | Poor aqueous solubility [87] | Soluble in water, acetone; slightly soluble in ether; insoluble in benzene [88] |
| Physicochemical property | White crystalline powder | White to yellowish crystalline powder |
| Melting point | 94 °C | 152 °C |
| pH Stability | Exhibits stability within a pH range of 3 to 10. | Unstable at pH levels greater than 7. |
| Light and heat stability | Durable under light and heat, resistant to oxidation | The light, heat, and metal ion stability of the substance is low, making it prone to oxidation [89] |

Based on a comparative stability study conducted by Tazesh et al. (2019) between KA and KDP under oxidative stress, it was observed that KDP underwent more rapid degradation in similar liquid oxidative stress conditions compared with KA [47]. This degradation could possibly be linked to the opening of the pyrone ring, followed by subsequent decomposition into smaller aliphatic chains. Based on the study, it was concluded that the notion of enhancing the stability of KA by obstructing its hydroxyl groups through the attachment of two palmitic acid molecules was a misconception, as the hydroxyl groups are not the reactive moiety of the molecule. However, Tazesh et al. (2019) still recommend choosing KDP over KA in cosmetic formulations. Yet, to prevent oxidation, formulators can include antioxidants to achieve improved stability results [52].

## 5. Mechanism of Action of Kojic Acid Dipalmitate

Kojic Acid Dipalmitate (KDP) demonstrates greater effectiveness compared with KA [89,90]. Esterases within skin cells hydrolyze KDP, leading to the in situ release of kojic acid, as illustrated in Figure 2 [34]. Consequently, the mechanism of action for KDP is akin to that of KA. The depigmentation properties of kojic acid, elucidated from cellular to molecular levels, have been extensively explored by Saeedi et al. (2019) [90].

Kojic acid, extensively studied as an inhibitor of tyrosinase, is recognized for its competitive inhibition of monophenolase activity and its mixed inhibitory effect on the diphenolase activity of mushroom tyrosinase [50]. Due to the action mechanism described, kojic acid is categorized as a "true inhibitor", wherein it can bind to the enzyme and inhibit tyrosinase activity. Tyrosinase is a copper-containing monooxygenase enzyme that catalyzes two reactions: o-hydroxylation of monophenols to catechols, also known as monophenolase or cresolase activity, and oxidation of catechols by $O_2$ to o-quinones, known as diphenolase or catecholase activity. Typically, true inhibitors are classified into four types: competitive inhibitors, uncompetitive inhibitors, mixed-type (competitive/uncompetitive) inhibitors, and non-competitive inhibitors (Figure 3). Competitive inhibitors are compounds that can bind to free enzymes, thereby preventing substrates from binding to the enzyme. The observed competitive inhibitory effect of kojic acid is attributed to its ability to chelate copper at the enzyme's active site. In contrast to competitive inhibitors, uncompetitive inhibitors can bind only to the enzyme–substrate complex. A mixed-type inhibitor, which

is both competitive and uncompetitive, can bind to both the free enzyme and the enzyme–substrate complex. Most mixed-type inhibitors bind to a free enzyme and an enzyme–substrate complex with the same equilibrium constant. In addition to the inhibitory mechanism, the strength of inhibition is a primary criterion for an inhibitor [50].

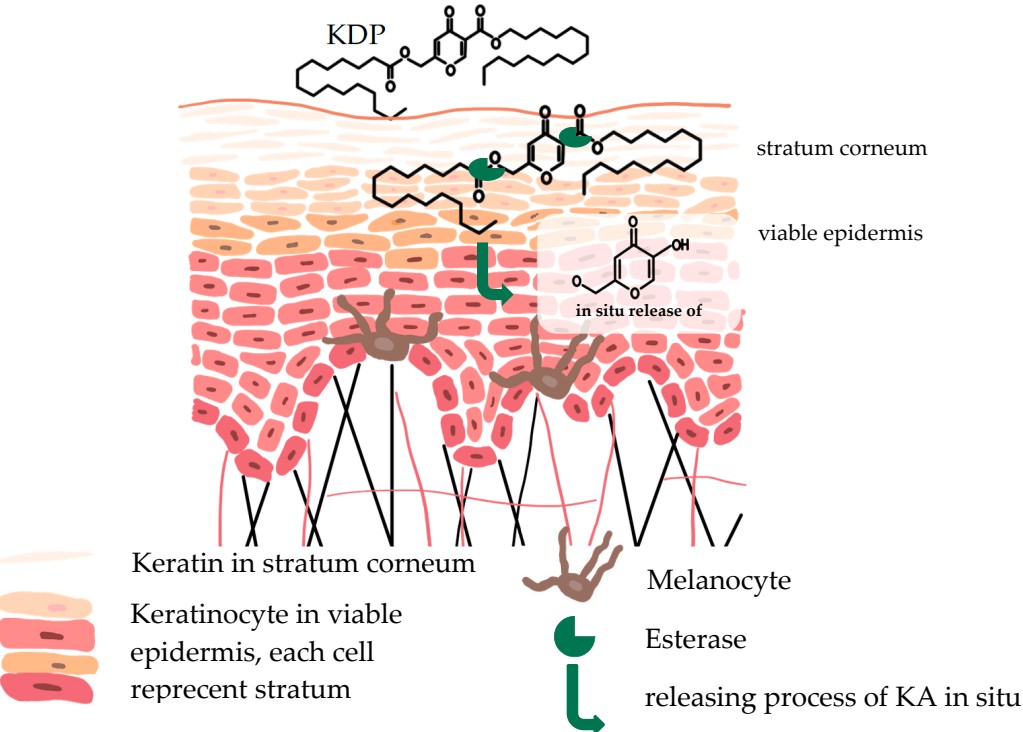

**Figure 2.** In situ liberation of KA. The enzymatic activity of esterases in the skin leads to the in situ liberation of kojic acid from KDP. This process takes place within the viable epidermis.

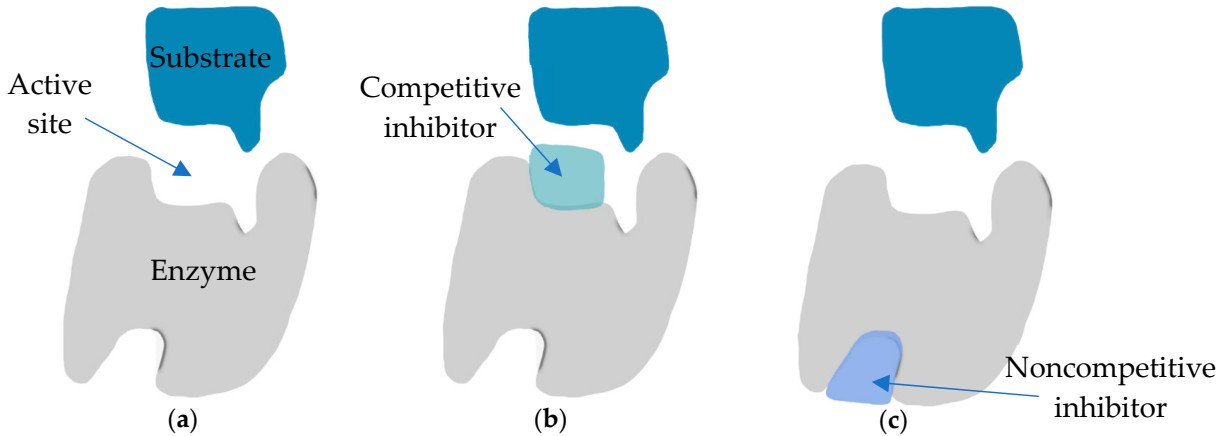

**Figure 3.** True inhibitor mechanisms of action: (**a**) normal binding; (**b**) competitive inhibition; (**c**) noncompetitive inhibition.

Casella et al. [91] studied how KA affects the oxidation of 3,5-dtbc (3,5-di-tert-butyl catechol) by dicopper model complexes. They suggested that KA acts as a connecting bridge between the metal centers in the dicopper(II) catalysts, indicating that KA may bind to the dicopper Ty center. This idea gained support from two other reports that characterized tetrachloro-o-catecholate-bridged dicopper(II) complexes [92]. Subsequent reports by Plenge et al. [93] and Ackermann et al. (2002) [94] put forward the concept of bridging and unsymmetric binding of catechol substrates in a Z2:Z1 fashion, with one of the two

oxygen atoms participating in a weak interaction with either of the neighboring copper(II) ions. Studies involving electron spin echoed envelope modulation (ESEEM) [95] and X-ray absorption spectroscopy (XAS) [96] of a met Ty-KA adduct from bacterial *Streptomyces antibioticus* Ty, providing further support for this binding mechanism. However, when the X-ray structure of the adduct of KA with the met form of Bacillus megaterium Ty was examined, it revealed that the KA molecule was situated at a distance of 7 Å from the dicopper center. This finding contrasted with the conclusions of Bochot et al. (2013) [97], who reported that the distances between copper and oxygen atoms of KA varied around 2.15 Å for CuB⋯O2, 2.04 Å for CuA⋯O2, and 2.17 Å for CuA⋯O3 [97].

Moreover, kojic acid has been reported as a slow-binding inhibitor of tyrosinase's diphenolase activity. Other potent slow-binding inhibitors of tyrosinase include tropolone and the substrate analog L-mimosine. Interestingly, all these slow-binding inhibitors of tyrosinase share a common feature: they contain an $\alpha$-hydroxyketone group. Kojic acid, tropolone, and L-mimosine are frequently used as positive controls in the literature to compare the inhibitory potency of newly discovered inhibitors [50].

## 6. Cosmetic Application of Kojic Acid Dipalmitate

### 6.1. Cosmetic Products Containing KDP

In skincare products, kojic acid dipalmitate was used at concentrations ranging from 0.01% to 25%. Typically, it was employed at concentrations between 0.2% and 8.0%, with the most frequent usage occurring at concentrations of 0.4% to 4.0% [98]. Kojic acid dipalmitate exhibited the ability to inhibit the activity of the tyrosinase enzyme, thereby decelerating melanin synthesis by impeding the conversion of DOPAchrome into DHICA [45].

A study conducted by Chandrashekar et al. in 2018 demonstrated that a 2% kojic acid dipalmitate formulation in a combination cream was effective and safe as a therapy for melasma. Kojic acid dipalmitate did not induce skin irritation and contributed to skin brightening, as evidenced by a reduction in hyperpigmentation observed in 51–57% of the subjects. Kojic acid at concentrations of 1–2% did not exhibit hepatocarcinogenic effects, was non-genotoxic, did not irritate the mucosal layer, and did not lead to sensitization [99,100].

As of the time when this literature review was conducted, it is known that there are over 132 cosmetics available on the market containing Kojic Acid Dipalmitate (KDP) in various formulations. These data were obtained from a list of products containing kojic acid dipalmitate accessed on the website [101]. These formulations include face or body creams, lotions, gels, face masks, serums, toners, eye-brightening products, lip products, face or body washes, and soap bars, as well as underarm creams designed to reduce pigmentation. All of these have been summarized in Figure 4.

The concentrations of KDP in these products typically range from 0.4% to 4.0%. KDP is most commonly formulated in creams designed for brightening both facial and body areas. In contrast, formulations containing KDP as face scrub and lip scrub are relatively rare. Additionally, KDP is incorporated into cosmetics intended for the whitening of the under-eye area and the lightening of the underarms. In addition to containing KDP, these whitening products may be provided as standalone treatments or in combination with other depigmenting agents such as arbutin, niacinamide, retinol, tranexamic acid (an antifibrinolytic agent widely favored in cosmetics for addressing melasma or hyperpigmentation), or in combination with exfoliant agents like glycolic acid and lactic acid. These products may also include antioxidants such as ascorbic acid and tocopherol acetate, as well as pine bark extracts [101].

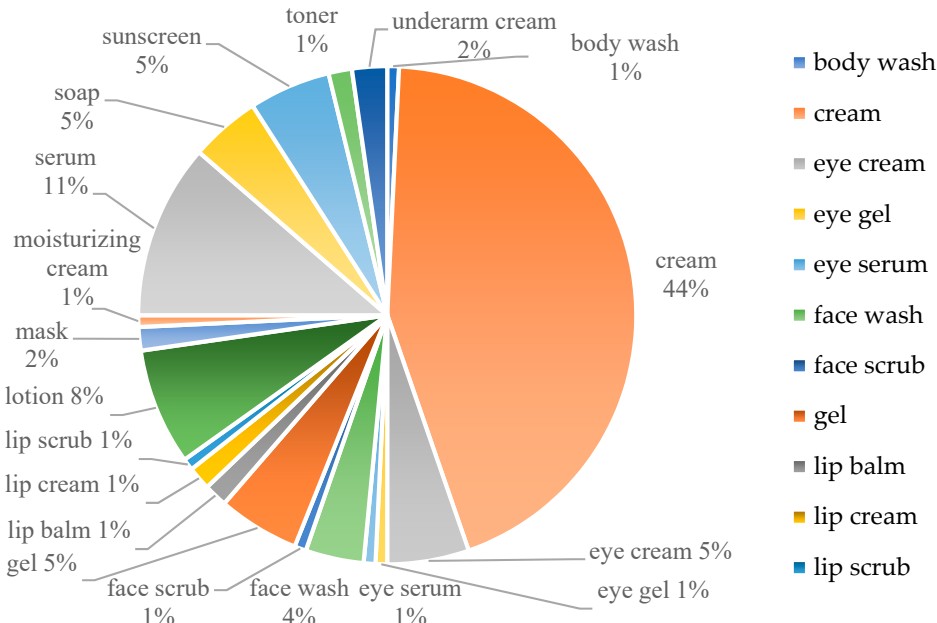

**Figure 4.** Cosmetics available in the market containing Kojic Acid Dipalmitate (KDP) in various formulations.

### 6.2. Patent Products of KDP

The patents related to kojic acid dipalmitate remain relatively limited. One such patent related to KDP is the US patent held by Whittemore et al. (1998), which claims a skin-whitening cosmetic composition containing kojic dipalmitate. The invention specifically pertains to an anhydrous skin-whitening cosmetic composition incorporating kojic dipalmitate [98].

The Shanghai Institute of Technology holds a patent for Solid Lipid Nanoparticles (SLN) containing kojic acid dipalmitate, valid until 2014. A disclosed innovation pertains to a nanometer-sized solid lipid carrier encapsulating kojic acid dipalmitate. This delivery system was designed to enhance the permeation of kojic acid dipalmitate into the skin, increase utilization efficiency, and improve the whitening effect components.

Components of the formulation shown in the patent include kojic acid dipalmitate, phospholipid, solid lipid, liquid lipid, a solid emulsifying agent, a polyalcohol additive, and a preservative, with the remaining content being deionized water. The method for preparing the nanometer solid lipid carrier involves melting an oil phase and ingredients that dissolved in it at 75–95 °C, heating a water phase containing deionized water and the polyalcohol additive, and combining the two phases. High-pressure homogenization is then performed, followed by stirring and cooling to produce the nanometer-sized solid lipid carrier [102].

As for the patent regarding the analytical approach of Kojic Acid Dipalmitate (KDP) using the High-Performance Liquid Chromatography (HPLC) method, it is held by Shanghai Jahwa United Co. Ltd., Shanghai, China [103]. The three patented formulations and characterizations containing kojic acid dipalmitate are presented in Table 2.

**Table 2.** Patents of kojic acid dipalmitate.

| Pantent Holder | Field of Invention | Year | No. of Patent | Reference |
|---|---|---|---|---|
| Jerry Whittemore Robert Neis, US | The present invention relates to a skin-whitening cosmetic composition and in particular to such a composition that is anhydrous and incorporates kojic dipalmitate. | 1998–2018 | US5824327A | [98] |
| Shanghai Institute of Technology, Shanghai, China | A kind of nano-solid lipid carrier and preparation method of coated kojic acid acid dipalmitate | 2014–2034 | CN104116643A | [102] |
| Shanghai Jahwa United Co Ltd., Shanghai, China | The present invention relates to a kind of high-performance liquid chromatography (HPLC) analytical approach, specifically related to a method with the HPLC quantitatively analyzing kojic dipalmitate. | 2002–2022 | CN1188700C | [103] |

## 7. Nanotechnology Formulations of Kojic Acid Dipalmitate

Nanotechnology refers to the manufacturing and use of materials at the nanoscale [104–107], where they exhibit distinct physicochemical characteristics compared with their larger particles [107]. These novel materials demonstrate an increased surface area as a result of certain internal rearrangements, leading to distinct interactions with biological systems [108]. The integration of nanotechnology into cosmetic formulations is regarded as the most current and developing technology currently accessible [105]. Cosmetic producers use nanoscale compounds to enhance UV protection [109–112], facilitate deeper skin penetration [113–117], prolong the effects [118,119], intensify color [120,121], improve finish quality [121], stability [118,122], and provide lower toxicity [87,115]. Kojic acid dipalmitate has been widely utilized in the cosmetics industry. It has been formulated using nanotechnology, primarily to enhance its physical and chemical properties.

Kojic acid dipalmitate incorporates two palmitate groups onto the hydroxyl group at C-7 [36], resulting in a molecular weight of KDP exceeding 500 Da, thereby impeding its permeability. As previously explained in the preceding section, the molecular weight of KDP is 618.9 g/mol [81]. According to the literature, a majority of chemical compounds with a molecular weight greater than 500 Da are unable to permeate the skin through passive diffusion processes [123]. To address this limitation, KDP has been formulated into various preparations such as nanosomes, nanocreams, multiple emulsions, liposomes, solid lipid nanoparticles (SLN), ethosomal suspensions, and nanoemulsions. These formulations aim to enhance skin permeability and stability, and reduce toxicity, thereby improving efficacy and conferring skin benefits. Figure 5 displays schematic representations of the architectures of KDP integrated into several types of nanomaterials, while Table 3 presents various cosmeceutical formulations of KDP utilizing nanotechnology.

**Table 3.** Enhanced cosmeceutical formulation of KDP.

| Published In | Preparation of KDP | The Research Objective | Diameter of Particle/Droplet | Zeta Potentials | Loading Capacity | Results | Reference |
|---|---|---|---|---|---|---|---|
| 2000 | Nanosome | Development of KDP nanosome in mono-vesicle and increased stability | 57–75.7 nm | −24 mV | NA | Turbidity was very good transparency compared with nanosome with liposome. It formed the monovesicle in the opposite direction to form the multi-lamellar vesicle of the liposome. The stability of nanosomes was very good for 6 months. | [124] |
| 2010 | Nanocream | Increased release and permeability through skin in vitro | <350 nm | NA | NA | Nanocreams had shown to produce a higher drug release and permeability through Franz diffusion cells, although there was no significant variation than that in normal cream at *p* value < 0.05. Nanocreams penetrate faster and the cumulative amount of KDP is higher than in normal creams. | [72] |
| 2015 | W/O/W Multiple Emulsions | Increased safety and activity of KDP in vitro | 0.056–12.487 μm | NA | N/A | Incorporation of KDP into MEs improved the safety and antioxidant activity of KDP in vitro. | [85] |
| 2020 | Liposome | Increasing stability and loading capacity | 80–100 nm; PDI ≤ 0.2 | −0.5 to −0.6 mV | 0.61% to 28.12% | Ethosomal gel had a good stability at lower temperature (8, 25 °C). KDP loading capacity increased from 0.61 to 28.12% | [125] |
| 2020 | Solid Lipid Nanoparticle (SLN) | Increase release profile and permeability through skin ex vivo | 70 nm | NA | 47% | The KDP loaded in the SLN presented a slower release profile of KDP in comparison with the formulations loaded with KDP. The KDP loaded into SLN had the highest concentration in the stratum corneum. | [48] |
| 2022 | Ethosomal suspension | Increase stability and skin benefits | 148 nm | −23.4 mV | 90.0008% | Ethosomal gel gave a significant decrease in skin melanin, erythema, and sebum levels while improving in skin hydration level and elasticity during non-invasive in vivo studies. The formulation had good stability at a lower temperature (8, 25 °C). | [87] |
| 2023 | Nanoemulsion | Increase permeation, antioxidant and depigmentation efficiency, and lower cytotoxicity | <130 nm | −10 mV | >95% | The nanoemulsion containing 1 mg/mL KDP exhibited antioxidant and depigmenting activities and allowed the active compound to reach the epidermis without permeating to deeper layers of the skin, showing potential for use in cosmetic formulations for melasma treatment. Such nanoemulsion was safe for fibroblast-like cells (3T3-L1) at concentrations up to 1%. | [126] |

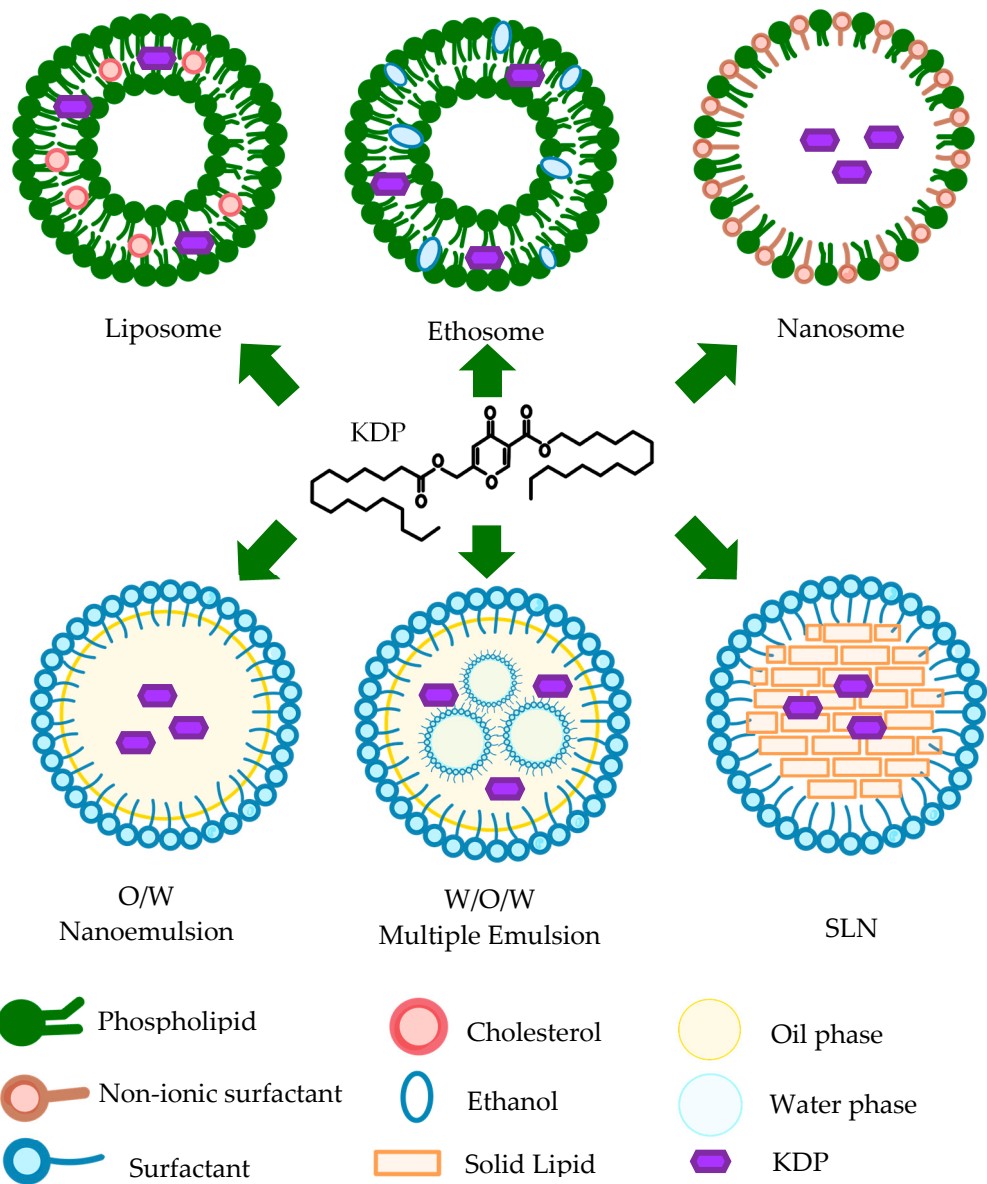

**Figure 5.** Schematic illustration of several configurations in which KDP nanomaterials were formulated.

### 7.1. Nanoemulsion

Similarly to KDP, many pharmaceutical ingredients are hydrophobic or have low solubility in water. Lipid-based delivery systems, such as nanoemulsified systems, are considered the optimal method for delivering medications that are water-soluble, insoluble, or weakly soluble due to their enhanced ability to solubilize and transport these pharmaceuticals [127–129]. Researchers have shown significant interest in nanoemulsions (NEs) for the development of diverse drug delivery systems in pharmaceutical applications via different administration routes [130–132].

Nanoemulsions (NEs) consist of droplets with diameters equal to or less than 100 nm [133], with an oil content varying from 5 to 20% $w/v$ [134]. Some specific NEs may use a combination of oils to enhance the solubility of drugs. To facilitate the stabilizing process, it is possible to include a co-surfactant/co-solvent in addition to the surfactant [135]. NEs exhibit distinct behavior compared with other emulsion systems, such as minimal agglomeration or precipitation, thereby reducing the likelihood of creaming or sedimentation. Thus, NEs exhibit greater stability compared with other emulsion systems [134].

Diverse methodologies and strategies are used for the fabrication of NEs, which can be categorized based on energy requirements, type of phase inversion, and self-emulsification. High-energy techniques are extensively used to generate NEs, which require more mechanical energy to generate powerful disruptive forces that transform bigger droplets into smaller ones [136]. Various mechanical equipment or instruments, such as ultrasonicators [137], microfluidizers [138–140], and high-pressure homogenizers [136], are used to create disruptive forces.

High-pressure valve homogenization (HPH) is a well-recognized and widely used technique for producing nanoemulsion [136]. August Gaulin created the basic technique in the early 1900s to decrease the size of fat globules in milk, with the goal of improving stability and prolonging shelf life [141]. This equipment operates based on a valve that is specifically built and paired with the use of high pressures. As a consequence, it creates a concentrated and confined area of extreme hydrodynamic stress. This process successfully divides the drops into smaller pieces, resulting in the generation of tiny drops that are required for the creation of nanoemulsions [142]. Microfluidization involves manipulating and controlling fluids at a small scale, often in the micrometer range [138–140]. Ultrasonication surpasses other high-energy technologies in terms of both operation and cleaning [137].

Emulsion production techniques that operate at low energy levels are more efficient due to their use of internal chemical energy inside the systems [134]. Phase inversion refers to the process of changing the polarity or composition of a substance [143]. Emulsification techniques involve the observation of phase transition occurring during the emulsification process, caused by the sudden change in curvature of the surfactant. The Phase Inversion Temperature (PIT) technique induces a quick change in the surfactant's curvature by temperature variations, resulting in the formulation of NEs [144]. The Phase Inversion Composition (PIC) approach achieves phase inversion by modifying the emulsion composition instead of adjusting the system temperature [145,146]. The Emulsion Inversion Phase (EIP) approach involves the inversion of emulsions via Coalescence and Phase Inversion (CPI) processes, affecting the proportion of the dispersed phase volume instead of the surfactant properties [147].

Kojic Acid Dipalmitate is one of the active ingredients that has been formulated in a nanoemulsion preparation using the high-energy method, employing the Ultra-Turax [126]. The study suggests that nanoemulsions serve as effective carriers, enhancing both the stability [148] and activity [72] of KDP on the skin. Beyond their carrier capabilities, nanoemulsions are recognized as a preferable option for drug delivery systems related to skin permeation. This preference arises from their diminished particle sizes and lipophilic characteristics [149], resulting in an increased affinity with the stratum corneum. Consequently, this facilitates deeper penetration and permeation of active substances into the skin, leading to heightened efficacy [150]. Additionally, nanoemulsions possess lipophilic cores, making them excellent carriers for hydrophobic actives in aqueous media [150].

*7.2. Nanocream*

Kojic acid dipalmitate (KDP) was also formulated in nanocream [148]. Nanocream is a formulation of nanoemulsion in the form of semisolid cream [151,152]. Nanoemulsions are composed of an isotropically clear dispersion of two liquids that are immiscible with each other, with droplet size varied between 100 and 200 nm [153,154]; in another study, 600 nm [151]. These liquids include a dispersed phase consisting of oils and a continuous phase consisting of water. The presence of dispersed-phase droplets that are smaller than 200 nanometers results in the nanoemulsion exhibiting a clear and transparent appearance [155]. The stability of this dispersion is maintained by an interfacial film of surfactant molecules, which possess stable thermodynamic properties [123,156].

Al-Edresi and Baie (2009) conducted a study aimed at formulating a nanoemulsion in cream form containing Kojic Acid Dipalmitate (KDP) as a whitening active ingredient and evaluating its stability [148]. The primary components of the nanocream formulation

included Emulium Kappa® (EK), which consists of candelilla/jojoba/rice bran polyglyceryl-3-esters, glyceryl stearate, stearoyl alcohol, and sodium stearoyl lactylate, serving as the emulsifier. Propylene glycol was used as a cosurfactant in a 9:1 ratio. The oil phase comprised virgin coconut oil (VCO) and squalene oil, maintaining a consistent surfactant-to-oil ratio of 1.4:1.2. The formulation of the nanoemulsion was successfully achieved using the Emulsion Inversion Point method.

To mitigate Ostwald ripening, the main instability mechanism of emulsion systems [149], insoluble oil (squalene)-was added to the system. The Ostwald ripening rate decreased significantly from 14.94 to 0.97 nm/day as the squalene concentration increased from 2 to 20%, representing a nearly 15-fold reduction in growth rate. This finding aligns with the study done by Cruz-Barrios (2014), which demonstrates that the inclusion of squalene in the formula mitigates the impact of ripening [157]. The zeta potential of the formulation indicated an increase in charge from $-65.1$ to $-101.8$ mV with the rising squalene ratio. This led to enhanced repulsion forces between the droplets, contributing to the improved stability of the nanoemulsion [158]. The negative droplet charge resulted from the adsorption of hydroxyl ions on the non-polar VCO droplet through hydrogen bonding [159]. The primary droplet diameter, ranging from 171.3 to 240.2 nm, remained unaffected by the squalene ratio. It is important to note that, while this research focused on enhancing the stability of the nanocream in terms of the zeta potential of the system, it did not assess the permeation of active ingredients into the skin [148]. In the subsequent development, Kojic acid dipalmitate (KDP) was further developed into an encapsulated form using phospholipids, namely, liposomes [125].

### 7.3. Liposome

Al-Edresi et al. (2020) conducted a study to enhance the loading capacity of kojic acid dipalmitate (KDP) into liposomes using the active loading method [125]. In this research, KDP was formulated in liposomes as encapsulating agents to overcome obstacles to cellular uptake [160–162] and target specific sites in vivo [163,164], thereby improving the delivery efficacy of compounds. Liposomal formulations were also proposed as a means of enhancing the therapeutic efficacy of poorly bioavailable drugs [165,166].

Initially, liposomes were prepared using the thin lipid film hydration method without active ingredients. To actively load KDP into the liposomes, a KDP solution was mixed with liposome suspension in a shaking water bath at 60 °C [167]. As the temperature of the liposomes increased to the phase transition temperature, holes opened in the lipid bilayer, allowing KDP to permeate from the intraliposomal to the interliposomal medium due to concentration gradients. This gradient served as a driving force for the permeation of KDP, leading to an equilibration of concentration on both sides of the liposome bilayer [168]. Active loading resulted in significantly higher loading capacity (%LC) compared with passive loading of KDP [163,168]. The concentration gradient technique forced KDP to be incorporated into the core of the liposomes, achieving an %LC of 28.12% [125].

Passive loading of KDP into liposomes, on the other hand, depended mainly on hydrophobic interaction and association with the 1,2-distearoyl-sn-glycero-3-phosphocholine (DSPC) bilayer structure as a phospholipid [168]. Liposomes were able to incorporate KDP, but in low percentages for passive loading methods, with a loading capacity of 0.61%, as the lipid bilayer constituted only small fractions of the liposomes.

Using the active loading method not only resulted in an increase in the amount of KDP incorporated into the liposomes but also maintained stable liposomes with particle sizes in the range of 80–100 nm, PDI ≤ 0.2, and zeta potential of $-0.5$ to $-0.6$ mV [125]. No significant changes in particle size over time were observed, indicating stable liposomes. However, it is important to note that this study did not conduct penetration testing into the skin or evaluate the content of formulations during storage stability testing.

### 7.4. Solid Lipid Nanoparticles

Solid lipid nanoparticles (SLNs), formerly referred to as liposperes, are a kind of pharmaceutical nanocarrier that show potential for controlled drug delivery [169,170]. SLNs are generally composed of biodegradable and safe lipidic components [171]. SLNs possess the notable ability to transport a wide range of therapeutic substances, such as tiny medication molecules, big biomacromolecules (such as polysaccharides), genetic material (such as DNA or siRNA), and vaccination antigens [172]. Small drug molecules have the ability to load both hydrophilic and lipophilic medicines, including KDP.

Kojic Acid Dipalmitate (KDP) have been formulated in SLN by Mohammadi et al., (2020) [48]. SLN-loaded KDP formulation consisted of melted GMS 100 mg and KDP 10 mg in 2% PVA using evaporated solvent ethanol/acetone 2.5:1.5, all of which had a KDP entrapment efficiency of about 47%, meaning the KDP concentration in the formulation was 10 mg/25 mL (0,04%) and being loaded in the SLN of about 47% of 0.04% (0.02% KDP in SLN). SLN-loaded KDP was successfully formulated with the mean size of $70 \pm 5$ nm. In this formulation, stability test results are not explained, so conclusions about the stability of the formula cannot be drawn. However, in vitro drug release and ex vivo permeation of Kojic Acid Dipalmitate (KDP) from Solid Lipid Nanoparticle (SLN)-based preparations were clearly described in the research paper.

The release profile of KDP from SLN preparations follows a first-order kinetic model. In comparison to KDP powder and KDP cream, formulations loaded with KDP in SLN, hydrogel, SLN-based cream, and SLN-based hydrogel exhibit a slower release rate. Among these, the KDP hydrogel demonstrates the slowest release profile, followed by the SLN-based hydrogel of KDP. These findings suggest that the lipophilic nature of KDP, the occlusive effect of the cream, the matrix structure of SLN, and the hydrogen bonds facilitated by polyvinyl alcohol (PVA) play crucial roles in determining the release rate of KDP and its diffusion into the receiving phase [48].

One of the factors accelerating the release rate from KDP powder and KDP cream is the lipophilic nature of KDP, which enables penetration into the skin through both intracellular and paracellular pathways. The second factor is the entrapment efficiency of KDP in the SLNs (47%). In contrast, KDP powder, KDP cream, and KDP hydrogel have higher concentrations, leading to a greater diffusion rate through the skin based on Fick's second law. The lipid matrix structure of SLNs retains lipophilic drugs for an extended period, allowing a slower release. The hydrogen bonds formed through the interaction of the hydrophilic structure of hydrogels, PVA in the SLNs, and even KDP itself contribute to a slower release rate from hydrogel formulations [48].

### 7.5. Ethosomes

Ethosomes are nanocarriers in the form of vesicles that contain a relatively high concentration of ethanol, ranging from 20% to 45% [123,173]. Ethanol is an effective substance that enhances permeation, giving ethosomes distinct characteristics, including high flexibility and deformability [174]. This enables them to profoundly enter the skin and boost the absorption and distribution of drugs. The augmented concentration of ethosomes offers significant benefits in delivering medicinal ingredients for many ailments, such as acne, psoriasis, alopecia, skin infections, hormone deficiencies [175], and hyperpigmentation [87].

The fundamental constituents of ethosomes consist of active pharmaceutical ingredients (API), ethanol, water, and phospholipids [176]. Ethosomal vesicles consist of a phospholipid bilayer around an aqueous core that holds the medication. Ethosomes differ from other lipid nanocarriers in terms of ethanol content, bilayer fluidity, absorption route via the skin, synthesis procedure, and absence of adverse effects. Ethosomes have a smooth and pliable size, ranging from 30 nm to several microns. It has been shown that ethosomes, although being manufactured using the same technique as liposomes, are smaller in size than liposomes [175]. The decrease in size is caused by the increased alcohol content, and it becomes progressively smaller as the percentage of ethanol increases to 20–45 [173].

Ethosomes have the ability to trap several types of molecules, including hydrophilic, lipophilic, and high-molecular-weight compounds [177]. Kojic Dipalmitate (KDP) can also be formulated in nanosized ethosome gel. Various ethosomal suspensions loaded with KDP were prepared using soy phosphatidylcholine, ethanol, propylene glycol, and water through a cold method. These formulations underwent assessment for size, zeta potential, polydispersity index, entrapment efficiency, FTIR spectroscopy, and scanning electron microscopy (SEM). Subsequently, the stability of the optimized gel was examined, and in vivo studies were conducted to evaluate the skin benefits. The optimized formulation has zeta potential, size, and entrapment efficiency of −23.4 mV, 148 nm, and 90.0008%, respectively. SEM results showed that the vesicles were spherical in shape. Ethosomal gel had good stability at lower temperatures (8, 25 °C). In addition, ethosomal gel causes a significant decrease in skin melanin, erythema, and sebum levels, while it causes improvement in skin hydration level and elasticity during non-invasive in vivo studies [87].

The overall findings indicated that the prepared KDP-loaded ethosomal formulation was stable and provided deep penetration of KDP into the skin. It offers a promising therapeutic approach for use in skin hyperpigmentation, as it has skin-whitening and moisturizing effects [87].

*7.6. Nanosome*

Nanosomes, also known as "nanoscaled liposomes", are tiny and homogenous microscopic vesicles made up of a phospholipid bilayer [178]. These vesicles may contain one or numerous lipid bilayers and have the capability to encapsulate pharmaceuticals. Nanometer-sized vesicles are composed of phospholipid bilayers, which may consist of a single or many lipid bilayers [179]. Therefore, they possess the desirable qualities of being harmless, non-stimulating to the immune system, and capable of being broken down naturally due to their resemblance to the molecular structure of mammalian cell membranes [180]. The nanoscaled liposomes possess comparable physicochemical and thermodynamic characteristics to those of regular liposomes. Several studies have shown that nanosomes exhibit greater encapsulation of nonpolar components compared with traditional liposomes due to their smaller size and better surface-to-volume ratio [134].

Lipid-based carriers can be prepared using various techniques, depending on factors such as solvent properties, drug release patterns, vesicles' size and uniformity, and potential toxicity. Some procedures for creating lipidic nanocarriers, known as "Nanosomes", include the supercritical fluid process (SCF) [181], microfluidization process [182], supercritical reverse phase evaporation method [183], and dual asymmetric centrifugation [178].

The supercritical fluid process (SCF) is used for manufacturing phospholipid nanosomes, which involves combining lipids and supercritical fluids under high pressure. The mixture is depressurized using a backpressure regulator, resulting in bubbles that dry out and form a lipid bilayer. This bilayer encloses solute molecules and seals itself, forming phospholipid nanosomes [181]. The microfluidization process allows for continuous and consistent formulation production, but requires high pressure, reaching up to 10 pounds per square inch (psi). Two types of microfluidized systems are used: single-step single-channel microfluidization and single-step dual-channel microfluidization [182]. The supercritical reverse phase evaporation method involves adding supercritical liquid carbon dioxide to a chamber, maintaining pressure above the supercritical threshold to ensure effective mixing. Once carbon dioxide evaporates, nanoliposomes with sizes ranging from 0.1 to 1.2 μm are formed [183]. Dual asymmetric centrifugation is a unique method that causes a vial containing a mixture of lipids and organic solvents to rotate on its vertical axis and spin around the centrifuge center, resulting in two simultaneous movements. This technique is not suitable for large-scale production due to its high encapsulation efficiency and high yield [178].

In 2000, In-young et al. introduced a novel encapsulation vesicle system that combines elements of both niosomes and liposomes, termed as nanosomes. Generally, when a surfactant is dissolved in water, it tends to form micelles [105,184]. A liposome is a

molecule with two lipophilic parts attached, such as a phospholipid [185]. Furthermore, when phospholipids and surfactants are mixed and dispersed in water, a monolayer is formed in a lamellar structure, as opposed to micelles. These monolayer vesicles are denoted as nanosomes. In comparison to liposomes, these vesicles exhibit a much finer size, contributing to enhanced stability of the active ingredient [124].

In this study, kojic acid dipalmitate was encapsulated inside the mono-layer vesicle and consisted of phospholipids and surfactants. The phospholipid used was hydrogenated liposomes (HL), and surfactants included in the formula were diethanolamine cetylphosphate (DEA-CP) and diglyceryl diodeate (DGDO). Kojic acid dipalmitate encapsulated in the vesicle could be up to 1% located in the core of the vesicles. With the application of the microfluidization (MF) method, the nanosomes were successfully developed until a nanosized suspension of the monovesicles system was obtained. It was confirmed through SEM that the particle size of the nanosomes was 57–75.7 nm, and the average particle size was 66 nm, indicating that a very fine particle size was formed. The stability of nanosomes developed in this research was also good, because they passed through MF three times, as confirmed by the zeta potential value at 23.8 mV [124].

### 7.7. Multiple Emulsion

Nanoemulsions have been recently acknowledged for their distinct features that render them more adaptable compared with conventional emulsion systems. In addition to the increasing interest in nanoemulsions, there have been notable advancements in the formulation of multiphase emulsions, which consist of droplets that contain immiscible droplets. Multiple emulsions serve as flexible platforms for chemically encapsulating components with varying polarity or solubilities, as well as for formulating multiphase materials and facilitating many additional tasks [186]. Widely employed processing tools, such as microfluidic devices and sequential emulsification, allow for accurate manipulation of the quantity, dimensions, and composition of the enclosed droplets [187]. This enables a wide range of possibilities for designing the internal structure of multiphase droplets and colloidal particles produced using these tools.

Kojic Dipalmitate (KDP) has also been formulated into multiple emulsions (MEs) with the aim of increasing their bioavailability and protecting the drugs against biological degradation and oxidation processes [86]. This formulation can extend the drug release, potentially reducing the required dosages and application time. The ME system formulated was in the form of a water-in-oil (W/O/W) system, developed through a two-step process. The initial W/O emulsion was first created using 20% span 80 as a surfactant, 45% liquid petrolatum, and 35% water. The primary emulsion was then dispersed into an aqueous solution of Tween 20 to generate a W/O/W ME composed of 80% of the primary emulsion, 10% of the solution in 40% Tween 20, and 10% water [86].

The droplet size of multiple emulsions (MEs) is notably larger in comparison with other nanodelivery systems, measuring approximately 1 μm with a zeta potential of −13 mV. In addition to the formulation, the authors conducted in vitro biological assays using the erythrocyte-induced hemolysis in vitro method to evaluate the potential irritation of a novel topical preparation. Free Kojic Acid Dipalmitate (KDP) led to the lysis of 4.09% ± 0.13% of erythrocyte membranes. KDP-unloaded MEs induced lysis of 1.57% ± 0.47% of erythrocyte membranes. The incorporation of KDP in ME resulted in 2.98% ± 1.12% lysis, demonstrating decreased erythrocyte lysis compared with free KDP. Therefore, all systems exhibited tolerable erythrocyte hemolysis [86].

The formulations, whether with or without the addition of KDP, underwent assessment for in vitro antioxidant activity over a 28-day period using the DPPH assay. Throughout the 28 days, there was a decline in the antioxidant power of all experimental groups, with the most significant decrease observed for free KDP. The differences between the samples were statistically significant ($p < 0.05$), and the observed lesser destabilization of the samples is likely attributed to the increased stabilization of the KDP-loaded ME formulation [86].

### 8. Conclusions

This article explores the development and use of Kojic Acid Dipalmitate (KDP) in skincare products, with a specific emphasis on its ability to hinder the production of melanin and its potential for treating disorders like melasma. KDP concentrations in skincare products often vary between 0.01% and 25% [98]. In 2018, Chandrashekar et al. conducted research that demonstrated that a combination cream including a 2% KDP formulation was both efficacious and safe for treating melasma [99,100].

The article explores the restricted number of patents associated with KDP, specifically focusing on one patent by Whittemore et al. (1998) for a cosmetic composition that lightens the skin and contains KDP [98]. Furthermore, the Shanghai Institute of Technology has a patent for Solid Lipid Nanoparticles (SLN) that includes KDP, with the purpose of augmenting skin permeability and enhancing whitening effects [102,103].

The fundamental focus of this discussion is the use of nanotechnology in the formulation of KDP. Various preparations, including nanoemulsions, nanocreams, liposomes, and ethosomal suspensions, are explored in detail. The purpose of these formulations is to overcome the molecular weight restrictions of KDP and improve the capacity of the substance to pass through the skin, as well as its stability and effectiveness, while also minimizing any potential harmful effects.

The article highlights the importance of nanoemulsions in drug delivery systems for skin permeation, providing evidence from studies that demonstrates their efficacy in improving both the stability and activity of KDP on the skin. Nanoemulsions are favored because of their smaller particle sizes and lipophilic properties, which enable enhanced penetration and absorption of active ingredients into the skin.

In addition, the essay provides a comprehensive analysis of formulations such as nanocreams and liposomes, specifically examining their stability, particle sizes, and loading capacities. The focus is on the creation of ethosomal suspensions loaded with KDP, which show potential for effectively treating skin hyperpigmentation.

Ultimately, the study highlights the transformative influence of nanotechnology on the development of Kojic Acid Dipalmitate, augmenting its physical and chemical characteristics. The use of KDP in several nanoscale formulations has significant promise for enhancing effectiveness and skin advantages in cosmetic uses.

### 9. Future Potential of Kojic Acid Dipalmitate

The future prospects of Kojic Acid Dipalmitate (KDP) in skincare products seem promising. Furthermore, the potential of KDP can be greatly enhanced by strategically combining it with other active ingredients. This can unlock synergistic effects that can significantly amplify its effectiveness in skincare formulations. Through the synergistic combination of KDP with complimentary active ingredients, the formulation is able to effectively address numerous facets of skin health, providing a holistic treatment for a range of skin issues, such as: (1) Enhanced Skin Brightening. Combining KDP with potent skin brightening agents such as arbutin or licorice extract may create a powerful synergy [23,188,189]. Arbutin, for example, is known for its melanin-inhibiting properties, and when combined with KDP, it may result in a more robust formula for addressing hyperpigmentation issues. (2) Antioxidant Protection. Incorporating antioxidants like vitamin C or green tea extract alongside KDP can provide dual benefits [190–193]. This combination not only inhibits melanin production but also shields the skin from oxidative stress, helping to prevent premature aging and promoting an overall healthier complexion; (3) Moisture Retention and Hydration. Formulating KDP with hyaluronic acid or glycerin can enhance the moisturizing properties of the skincare product [107,194]. This combination ensures that the skin not only receives the benefits of melanin inhibition but also maintains optimal hydration, contributing to a more radiant and supple complexion; (4) Anti-Inflammatory Support. Partnering KDP with anti-inflammatory agents such as chamomile extract or aloe vera can be beneficial, especially for individuals with sensitive skin [24,195]. This combination potentially mitigates potential irritations and redness, making the skincare product

suitable for a broader range of skin types; (5) Collagen Boosting. Introducing collagen-boosting ingredients like peptides or retinol alongside KDP can support skin elasticity and firmness [188,196]. This combination potentially addresses not only pigmentation concerns but also contributes to a more youthful and resilient skin appearance; (6) Sunscreen Integration. Combining KDP with a broad-spectrum sunscreen offers a comprehensive approach to skin protection. Sunscreen ingredients like zinc oxide or titanium dioxide can complement KDP by preventing UV-induced pigmentation, providing a well-rounded defense against sun-related skin issues; and (7) Customizable Formulations. Considering individual skin needs, formulating KDP with ingredients tailored to specific concerns, such as acne-fighting agents (e.g., salicylic acid) [197] or anti-aging compounds (e.g., peptides), may potentially create personalized skincare solutions.

In conclusion, the future potential of Kojic Acid Dipalmitate lies in its ability to synergize with a spectrum of other active ingredients. These combinations have the potential to create advanced skincare formulations that not only target pigmentation issues but also offer a holistic approach to skin health, catering to diverse needs and preferences in the ever-evolving landscape of skincare.

**Author Contributions:** Conceptualization, A.Y.C. and I.S.K.S.; methodology, A.A., A.Y.C., I.S.K.S. and S.M.; data curation, A.A. and A.Y.C.; writing—original draft preparation, A.A.; writing—review and editing, A.Y.C. and A.A.; supervision, A.Y.C., I.S.K.S. and S.M.; funding acquisition, A.Y.C. All authors have read and agreed to the published version of the manuscript.

**Funding:** The Ministry of Health, Republic of Indonesia, provide funding for this research, while Padjajaran University funded the APC.

**Institutional Review Board Statement:** Not available.

**Informed Consent Statement:** Not applicable.

**Data Availability Statement:** Not available.

**Conflicts of Interest:** The authors declare no conflict of interest.

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
