# Peer review of "Nanotechnology-Enhanced Cosmetic Application of Kojic Acid Dipalmitate, a Kojic Acid Derivate with Improved Properties"

_cosmetics, doi:10.3390/cosmetics11010021_

Round 1

Reviewer 1 Report

Comments and Suggestions for Authors

The review presented by the authors is relevant in the field. Here, I give my suggestions and comments.

1.    The title does not correspond to the emphasis of the information presented. I suggest changing the title to the most relevant information (nanoformulations).

2.    The authors are giving too much importance to the nanoformulations, and this is not stated in the title, abstract, and introduction sections. I suggest including this information in such sections.

3.    In Section 2. Sythesis of Kojic Acid Dipalmitate: There is no correct reference to Figure 1. Did the authors create the figure? Which program or app did they use? Is it a modification of another article?

4.    In Section 3. I suggest including a table to summarize the described properties of KDP.

5.    In section 4, Figure 2 must be improved to make it clearer. The figure legend does not include information about the green arrows. What do they indicate? Besides, there is no reference to the figure, the same as comment number 3. Please include the way you created the image.

6.    I suggest including more recent references. For example: Lokman Hakim, N. Y. D., Joginder Singh, H. K., Kang Nien, H., Siau Hui, M., & Zee Wei, L. (2023). Kojic Acid and Kojic Acid Ester: Review on Nanotechnology-based Approach for Enhancing the Delivery Efficacy. Recent Advances in Drug Delivery and Formulation: Formerly Recent Patents on Drug Delivery & Formulation17(2), 90-101.

Comments on the Quality of English Language

Moderate editing of English language required

Reviewer 2 Report

Comments and Suggestions for Authors

All of the comments are attached as a file.

Need more relevant citations since certain subtopics do not meet enough review.

The title itself shows the comprehensive review, but lack somewhere in the manuscript.

Round 2

Reviewer 1 Report

Comments and Suggestions for Authors

The authors have addressed the suggested changes to the review, the presented information is clearer and better organized. I consider the work will be helpful for the community in the field

Comments on the Quality of English Language

I suggest reviewing the minor mistakes in the English language by inviting a native speaker to edit the manuscript. In general, the English language is well used.

Reviewer 2 Report

Comments and Suggestions for Authors

The revision has been done intensively. All of the previous comments are revised.